# Assessing the Performance and Participation among Young Male and Female Entrepreneurs in Agribusiness: A Case Study of the Rice and Maize Subsectors in Cameroon

**Djomo Choumbou Raoul Fani** [1,*], **Ukpe Udeme Henrietta** [2], **Emmanuel Njock Oben** [1], **Donald Denen Dzever** [3], **Onyeje Hephzibah Obekpa** [3], **Auguste Tamba Nde** [4], **Mohamadou Sani** [3], **Mbong Grace Annih** [5,6] **and Dontsop Nguezet Paul Martin** [7]

1    Department of Agricultural Economics and Agribusiness, University of Buea, Buea P.O. Box 63, Cameroon; obenjock@gmail.com
2    Department of Agricultural Economics and Extension, Federal University, Wukari, Wukari P.MB. 1020, Taraba State, Nigeria; ukpe@fuwukari.edu.ng
3    Department of Agricultural Economics, University of Agriculture, Makurdi, Makurdi P.M.B. 2373, Benue State, Nigeria; dzeverdenen@gmail.com (D.D.D.); obekpahephzibah@gmail.com (O.H.O.); sanimohamadou47@gmail.com (M.S.)
4    Department of Agricultural Economics and Extension, Bayero University, Kano P.M.B. 3011, Kano State, Nigeria; ndearnoul@gmail.com
5    Department of Crop Production Technology, University of Bamenda, Bambili P.O. Box 39, Cameroon; gracembong@yahoo.com
6    Ministry of Agriculture and Rural Development, Yaoundé P.O. Box 1639, Cameroon
7    International Institute of Tropical Agriculture, Bukavu 0970 DR, Congo; p.dontsop@cgiar.org
*    Correspondence: djomo.choumbou@ubuea.cm

**Abstract:** The role played by youth in agriculture cannot be overemphasized, while agribusinesses are important generators of employment and income worldwide. Improving the sustainability of food value chains can benefit millions of rural poor people living in developing countries, especially young women. Despite policies and programs aimed at encouraging agricultural entrepreneurs, they are still facing challenges such as high-cost agrochemicals, limited access to credit, price volatility, etc. which seriously affect their capacity to remain in the sector. This study was undertaken to assess the performance and participation of young male and female entrepreneurs. This study made use of cross-section data from a survey conducted on 1019 young agricultural entrepreneurs in Cameroon. The data were analyzed using gross margin, student t-test, and logistic regression. Our study contributes to the literature by showing that young women agribusiness entrepreneurs are as competitive as their male counterparts and that the opportunities for young women in agriculture are good. Incentives such as single-digit interest rates and no collateral security should be directed to young women to receive more credit for purchasing agrochemicals and improved varieties of seed. Furthermore, a price control policy should be set up throughout the year to enable young women earn as their young men counterparts to enable them remain in production and marketing activities.

**Keywords:** performance; participation; youth; entrepreneurs; male; female; gross margin; maize; rice

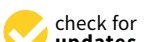



## 1. Introduction

Africa is experiencing rapid social and economic growth. Many economies are growing by more than 6 percent a year; however, large disparities in income distribution persist both within and among countries. As a result, rural households seek to escape poverty by engaging in market-oriented farming but with limited success due to a lack of innovative solutions to production and marketing constraints [1].

Agribusiness encompasses a wide range of activities that generate economic value. Agribusiness is comprised of not only farming but also all the other industries and services that connect farmers to consumers. It is traditionally thought that as an economy develops,

the role of agriculture in the economy's GDP and employment rates decreases. This trend has been observed in sub-Saharan Africa where agriculture's contribution to GDP has fallen from 43 percent in 1965 to 12 percent in 2008. If agricultural activity continues to be limited to crop and livestock production, it will fail to contribute to output growth and poverty reduction [2]. However, agriculture's contribution could be significantly enhanced by strengthening linkages with industry through agro-processing and providing value-addition to agricultural products, as well as improving post-harvest operations, storage, distribution, and logistics. Such an agribusiness development path paves the way for economic growth, structural transformation, and improved technical skills which in turn can catalyze economic activities and connect major economic sectors [2].

Entrepreneurial activities play a critical role in the development and well-being of societies [3,4]. Thus, various stakeholders including governments, non-profit organizations, researchers, and individuals are interested in facilitating the development of supportive entrepreneurial ecosystems. However, the growth of young female entrepreneurship has lagged behind those of men in many developed and in most developing countries. Understanding potential roadblocks that female entrepreneurs face is important for increasing their participation in entrepreneurial activity [4].

Cameroon's agribusiness practice operates in a market economy way. Young entrepreneurs who see opportunities to compete and thrive must save and invest on their own—or in others'—farms or businesses with a target to sell their output to earn income and pay employees. It is also important to remember that paid employment is both a business cost and the foundation of business revenue. They invest in farm business with the expectation to get a return or to breakeven. In case of no gain, there is a high tendency to move from farming to rural and urban migration. So, it is important to document the actual outcomes and opportunities for women as agribusiness entrepreneurs [5].

In Cameroon, maize was mainly considered a food crop in past decades and not a cash crop [6]. However, with high demands from brewery companies and the livestock sector, production has increased its importance as a cash crop. Maize production has increased drastically from 650,000 tons (t) in 2000 to 1,647,036 t in 2013. Rice has become the most rapidly growing food source for millions of people in Cameroon [6]. Accordingly, the Government of Cameroon established three development companies [Société d' Expansion et de Modernisation de Riziculture de Yagoua (SEMRY) in 1954; Upper Noun Valley Development Authority (UNVDA) in 1974; and Société de Dévelopement de la Riziculture dans la plaine de Mbo (SODERIM) in 1978] to boost rice production and the farmers' ability to increase their earnings (profitability). Despite these efforts, Cameroon produces an estimated 80,000 t of rice annually which is far short of the over 500,000 t required to meet national demand [7,8].

Attracting youth to and retaining them in the agricultural sector remains a global challenge. Many developing countries are faced with the challenge of ensuring food security for their growing populations amidst a decline in youth engagement in agriculture [9,10]. A study carried out in Reference [11] showed that youths working in agriculture mainly operate in small, unincorporated family businesses as self-employees or as contributing family workers without pay. While the employment opportunities available in the sector continue to increase for graduates in agriculture, in many countries, too few youths have embraced food production as a career path [9].

Agriculture plays a central role in the provision of productive employment for Africa's youth [12,13]. The reason is that it is the only sector that has the potential to provide the number of jobs required in the short term; moreover, it is argued that youth engagement in the sector will help counteract an aging farming population and make a positive contribution to food security and sustainability. However, if the sector is to be the sweet spot for youth employment, it must become more attractive, more productive, and more profitable, it must modernize and be less laborious. It is here that the new-found interest in youth, with its particular emphasis on rural entrepreneurship, skills enhancement, innovation,

and value chains, meets the long-standing concerns of agricultural and rural development, including technological change, extension, land reform, infrastructure, and markets [13].

It is difficult to give an accurate assessment of women's contributions to agriculture in Cameroon, due to a lack of data, it is estimated that rural women provide approximately 90 percent of the food needed for the sustenace of the population [14]. When it comes to the subject of gender under-representation in agricultural production and productivity, African Development Bank [15] affirmed that the African continent's diversity has prompted the various roles assigned to male and female gender. Gender disparities in access, use, and competitiveness of resources in agricultural production have been critical challenges for achieving food security and inclusive growth in Nigeria and Africa at large [16]. Different studies have looked into women's role in production in Africa, but few have looked into their roles and presence in Agribusiness in Africa.

Women may face greater difficulties in entering competitive markets because men might take over crop production from them once it becomes viable [17]. Much of the literature on sub-Saharan Africa found that women have traditionally been excluded from contractual agreements and agribusiness with private investors owing to their restricted direct access to land and ownership of economic capital by contrasting male and female household participation in agribusiness [18]. Various studies on this subject matter has been carried out in other continents and Africa, some of these works include [16] using Nigeria as a case study, [19] using Argentina as a case study, [20] which looked at women's participation in agribusiness in Brazil but no study investigating the participation of women in agribusiness in Cameroon has been carried out, a gap which necessitated this study.

In the past, youth were too seldom considered as a separate and pivotal interest group within the rural transformation. Rather, it was assumed that what must be good for rural communities as a whole is necessarily beneficial to youth, and at best their interests were linked to those of women and other disadvantaged groups. This trend is now being challenged where youth, especially educated youth returning to rural areas, are viewed as a key entry point for new agribusiness which results in employment creation. Indeed, rural youth are our future farmers and are most likely to adopt modern farming and agribusiness methods, assume market orientation, and ready to fill the current vacuum in the provision of services and logistics that are essential in the overall development of agriculture and agribusiness [1].

It is against this background that this study assessed the performance and participation among young entrepreneurs in agribusiness in order to fill the knowledge gap and empirically document policy for decision-making, which will encourage youths especially women to become agribusiness entrepreneurs. A long-term objective is to show the decision-makers that gender-blind policies are needed, but gender-positive information may be needed so that public investment in agrochemical, grain marketing, and roads can and will be put to very productive and helpful use for young farmers. A consequent research hypothesis is stated as follow:

**Hypothesis 1 (H1).** *Is there a significant difference between the profit among young male and female entrepreneurs in maize and rice agribusiness?*

The paper is structured as follows: Section 1 presents a brief/research hypothesis. Section 2 shows the literature review. Section 3 elaborates the methodology which includes the study area, method of data collection, population and sampling techniques, techniques of data analysis, validity and reliability of data, and models specification. Section 4 presents the results and discussion; the last section concludes the article.

## 2. Literature Review

Our literature development focused on gender and entrepreneurship. We contend that gender and entrepreneurship could be assessed in the following ways: economic and non-economic outcomes [21]. Studies in entrepreneurship have invoked a variety of theoretical perspectives to explain differences between female-owned businesses and male-owned businesses [22,23].

### 2.1. Non-Economic Outcomes

Following [21] non-economic outcomes such as self-empowerment, time flexibility, status in the community, satisfaction with life, and work–life balance are more important for women than for men. As we discuss below, the narrow definition of success that highlights only economic motivations for entering entrepreneurship tends to better fit the male model. Women often have different motivations when they enter self-employment and they evaluate the success of their business using different metrics than men. Thus, in discussing female entrepreneurial performance, it is important to include non-economic outcomes, which are frequently the driving forces behind women's choice of self-employment. However, the literature on non-economic business outcomes of women entrepreneurs is very sparse. There is some limited evidence that in evaluating their firm's performance, women tend to focus more on non-economic factors, such as personal fulfillment, flexibility, and desire to serve the community [24]. A study in Sweden [25] found that while women entrepreneurs were similar to men in their pursuit of economic goals, women also valued other goals, including customer satisfaction and personal flexibility. In a study of Lebanese female entrepreneurs [26], it was found that many women named non-financial aspects of their businesses, such as love of what they do every day and rendering an important service to the community, as important satisfying factors. A study of U.S. entrepreneurs revealed that women were more likely than men to develop strategies that emphasized product quality and less likely to emphasize cost efficiency [27].

### 2.2. Economic Outcomes

Many studies found that female-owned enterprises exhibit lower profitability and productivity than male-owned ones. The differences vary widely across studies [21]. Another study [28] found that female-owned firms throughout Latin America tend to be smaller than male-owned firms in terms of sales and number of employees. Similarly, Reference [29] found that female-owned firms in sub-Saharan Africa have sales that are 31 percent lower than male-owned firms. There could be many reasons why female-run businesses are smaller. In sub-Saharan Africa, Reference [30] estimates the gender gaps in labor productivity to be 12 percent. Reference [31] analyzes rural non-farm entrepreneurship in Ethiopia and finds that male-owned firms are three times more productive than female-owned ones. Some of the differences in performance can be explained by the type of firms women operate. In particular, the size and sector of the firm often explain a large portion of the differences in performance. For example, in the U.S., Reference [32] estimates that women's concentration in the personal service sector explains as much as 14 percent of the earnings differential. Reference [33] found that once differences in the sector are accounted for, there is no longer a significant difference in performance between male and female-owned businesses. However, many studies found that even after controlling for firm characteristics, there are still differences in performance. For example, in Madagascar, Reference [34] found that the estimated gender performance gap in value added is 28 percent even after controlling for factor inputs endowment, sectors, and the owner's human capital. In sub-Saharan Africa, only about one-third of the productivity gap is explained by differences in the types of businesses women run; smaller firms, firms that are unaffiliated with other businesses, and firms that are not registered [30]. In Uganda, a small sample mixed-methods study in Reference [35] found that when women cross over into male-dominated sectors, they attain higher returns than women in female-dominated sectors. In other words, the returns in male-dominated sectors are high not only for men. Even if

women get lower profits than men, they are still making more profits in male-dominated sectors than they would in female-dominated sectors.

An important aspect of performance evaluation that received little attention is the risk-return trade-off. Because women tend to be more risk-averse than men, they may choose to focus on lower risk/lower return strategies rather than high-risk/high-return strategies. References [21,36] found that although profits are significantly higher for male-controlled small and micro enterprises, so is the risk (i.e., the variation in profits). Reference [37] argues that it is inappropriate to compare returns from these different types of businesses without considering the differences in risk. They posit that inadequate control for differences in risk may explain why most of the previous literature observed differences in performance between male and female businesses. Indeed, when they control for risk-adjusted returns using the Sharpe ratio, measured as the ratio of profits over the variance of the profits, they find no differences in performance in their U.S. sample.

## 3. Materials and Methods

### 3.1. Research Philosophy, Research Design, and Research Approach

This study adopts an analytic form of a survey that makes use of cross-sectional data. This approach aims at determining the performance of young entrepreneurs engaged in maize and rice agribusiness. This approach examines the relationship between the revenue and cost incurred by young entrepreneurs as they exist in a defined population at a single point in time or over a short period of time.

### 3.2. The Study Area

Two challenges and an opportunity motivated this study in Cameroon. The challenges are rural un- and underemployment and poverty. In 2014, young women were seventy percent (467,700 young women) of the youths aged 15–24 years in Cameroon who were not in employment, education, or training, and a third of all households were officially poor with rural underemployment and poverty despite the fact that farming accounts for 70 percent of the workforce but contributing for just 17 percent of GDP [38]. Half a million young women and a quarter-million young men are ready and willing to grow food, make goods, and provide services to others. It is a huge resource; how will they be mobilized? These attributes make Cameroon suitable for the study. This study was conducted in Cameroon; the country has ten regions, namely, Centre; Littoral; Adamawa; Far-North; North; South; East; West; North-West, and South-West. It covers a total land area of 475,442sq km and is located in the Central part of Africa within latitudes 2° and 13° North and longitude 9° and 16° East of the equator [39].

### 3.3. Validity and Reliability of Instruments

The research instrument was validated by pilot testing and bypassing it through my supervisors to ensure that it possesses both face and content validity. The reliability of the instrument was conducted using a test–retest method. In doing this, twenty (20) copies of the questionnaires were administered twice to two communities drawn from the sample frame within the interval of two weeks. The scores obtained were correlated using Pearson Product moment correlation coefficient(r) for scores obtained at the interval level, while the Spearman's rank (rho) correlation was used for scores obtained at the ordinal level. A mean product–moment correlation coefficient (r) of 0.83 indicated high reliability.

### 3.4. Population and Sampling Techniques

A multi-stage, stratified simple random sampling technique was adopted to collect data used in the study with the help of well-structured questionnaires. In the first stage, three out of the ten regions in Cameroon were purposively selected based on the *a priori* knowledge that those regions are maize and rice-producing areas. In the second stage, two major markets for maize and rice were randomly selected in each of the three regions, and a total of six markets were selected in West, North, and Far North. In the third stage, a

list of maize and rice farmers was obtained from the regional delegation of the ministry of agriculture and rural development. From a total sample frame of 2223 and 1615 maize and rice producers and marketers, respectively, were stratified into young males and females following natural criteria such as education, age, and membership into association. The sample sizes of the various strata were obtained by randomization to get the number of respondents for the various strata; given that the population size of rice and maize producers and marketers are well known, a 5 percent error margin was adopted to select the sample size. By application of the Taro Yamane formula given below, rice and maize producers and marketers were randomly selected from each group which gave a sample size of 428 (288 males and 140 females) and 591 (434 males and 157 females), respectively.

$$n = \frac{N}{(1 + Ne^2)} \tag{1}$$

$n$ = sample size;
$N$ = population of Rice or Maize Producer and Marketers
$e$ = error margin

*3.5. Techniques of Data Analysis*

Given the smallholding and scattered nature of farms, Gross margin analysis and logistic regression were used to assess the performance and the determinants of participation, respectively. T-test was used to test whether there is a significant difference in the performance (proxy to profitability) between young males and females.

*3.6. Models Specification*

3.6.1. Gross Margin

Gross margin is one of the oldest and simplest analytical tools used in farm management. It has been used in a number of economic studies for analyzing the profitability of farm practice where fixed costs such as machinery, buildings, and other implements are not accountable [40].

Following [40] Gross Margin is given as:

$$
\begin{aligned}
\textbf{Gross margin} \quad = \quad & \text{Selling Price} \times \text{Quantity of output sold} \\
& - (\text{cost of farm labour } + \text{ cost of seeds } + \text{ cost of fertilizer} \\
& + \text{ cost of herbicides } + \text{ Transportation Cost } + \text{ Tax Paid} \\
& + \text{ Postharvest processing } + \text{ cost of chemicals} \\
& + \text{ drying fee } + \text{cost of packaging } + \text{Storage Cost})
\end{aligned} \tag{2}
$$

where Gross Margin is measured in FCFA/kg (1 FCFA = 0.0017 USD); selling price is measured in FCFA, it is a price at which producers and marketers sell their produce (rice and maize) in the market. It is used because agricultural price fluctuations determine the interaction between supply and demand forces [41]. Quantity of output is measured in kg; it is the quantity of rice or maize sold by a producer-marketer. It is among the common factors affecting the income of farmers [42]. The cost of farm labor is measured in FCFA. It is the cost incurred by rice and maize producers and marketers per laborer per day for activities such as clearing, planting, weeding, agrochemical application, and harvesting. Cost of seeds is measured in FCFA; seeds are the primary input used by farmers, contributing at least 40% to the formation of crop yields [43]. Cost of artificial fertilizer is measured in FCFA; artificial fertilizer is a substance containing the chemical elements that improve the growth and productiveness of plants. The cost of herbicides is measured in FCFA. Their use is due to the availability of low-cost herbicides and because of a similar shortage of manpower [44]. Postharvest processing is measured in FCFA. It is the sum of the quantity of outputs lost in operations such as threshing, transportation, processing, storage, and exchange before they reach the consumer. The cost of chemicals is measured in FCFA. It represents the chemicals used for the preservation of rice and maize

after the harvest. The drying fee is measured in FCFA. It is the amount paid off-farm to laborers because grain to be stored through warm weather needs to be dried, but energy is needed to remove moisture. The cost of packaging is measured in FCFA. The packaging is an essential part of a long-term incremental development process to reduce losses [45]. Transportation cost is measured in FCFA; it is the cost incurred by rice and maize marketers by moving their goods to market. Tax paid is measured in FCFA; it is a mandatory charge for everyone involved in income-generating activities in Cameroon. Storage cost is the amount paid daily by rice and maize marketers for keeping their goods in a warehouse.

### 3.6.2. T-Test Analysis

It was used to determine whether there is a significant difference between the means of the two sets of samples (young male and young female) drawn from the same sample frame. The respondents were tested under the same period.

The *t*- statistic to test whether the means (gross margins) are different will be calculated as:

$$t = \frac{(X_1 - X_2)}{\sqrt{(S^2_1/n_1) + (S^2_2/n_2)}}$$

where;

$X_1$ = mean value for credit users
$X_2$ = mean value for non-credit users.
$S^2_1$ and $S^2_2$ are the sample variance for sample $n_1$ and $n_2$, respectively.
T, follows the distribution with $n_1 + n_2$—2 degrees of freedom.

### 3.6.3. Logistic Regression

Following Reference [46], logistic regression can be specified as follow:

$$\begin{aligned} Y = {} & \beta0 + \beta1(\text{Age}) + \beta2(\text{Household size}) + \beta3(\text{Experience}) + \beta4(\text{Membership to association}) \\ & + \beta5(\text{Years in school}) + \beta6(\text{farm size}) + \beta7(\text{monthly income}) + \beta8(\text{price of seed}) \\ & + \beta9(\text{price of fertilizer}) + \beta10(\text{price of herbicides}) + \beta11(\text{selling price}) \\ & + \beta12(\text{distance to market}) + \varepsilon \end{aligned}$$

Y = 1 if the ratio sales/consumption is greater than 1and Y = 0 otherwise)
Age is measured in years.
Household size is measured in the number of people living in a household.
Experience is measured in years.
Membership in an association is measured as a dummy variable (member = 1; non-member = 0).
Years in school are measured in the number of years spent on formal education.
Farm size is measured in ha and determines the level of productivity.
Monthly income is measured in FCFA. It determines the capacity of the two researched groups.
Cost of seeds is measured in FCFA.
Cost of fertilizer is measured in FCFA.
Cost of herbicides is measured in FCFA.
Selling price is measured in FCFA.
Distance to market is measured in km. It could increase market access to customers.
$\varepsilon$ is the error term.

## 4. Results and Discussion

### 4.1. Strengths and Weaknesses of Models Used in This Study

Gross margin has been used in a number of economic studies for analyzing the profitability of farm practice where fixed costs such as machinery, buildings, and other implements are not accountable. However, gross margin measures only the profitability of the firm and ignores other factors such as an increase in the cost of production to secure a supplier or a decrease in the selling price to increase market share, etc. Gross

profit may produce misleading figures of profit. In logistic regression, outputs have a nice probabilistic interpretation, and the algorithm can be regularized to avoid over-fitting. Logistic models can be updated easily with new data using stochastic gradient descent. However, logistic regression tends to underperform when there are multiple or non-linear decision boundaries.

*4.2. Performance (Proxy to Profitability) among Young Male and Female Rice and Maize Producers and Marketers*

The result of profitability in rice for both researched groups presented in Table 1 shows a low average return for the amount of FCFA invested though higher for the females than the counterpart males. Furthermore, results of profitability in maize for both researched groups show a low average return for FCFA invested, though higher for males than for females as shown in Figure 1. The more investments male farmers made, the higher their profits compare to their female counterparts.This implies that the low return for investment could be due to the lack of subsidies in agrochemicals, the limited amount of credit received, selling price, and rural–urban migration. This could also be explained by the fact that in smallholding farming, there is a tendency to spend more on agrochemicals and intensive use of labor to increase farm profit. This result could further be explained by Reference [36] who found that although profits are significantly higher for male-controlled small and micro-enterprises than female-controlled small and micro-enterprises due to risk (i.e., the variation in profits). Moreover, Reference [34] found that while returns to capital in female-owned firms are significantly lower than in male-owned firms, the returns to time and hours of labor are actually higher for female-run firms. This suggests that economic outcomes can be improved if women could devote more time to their business ventures.

Over four-fifths (82 percent) of the female maize farmers, we surveyed paid at least some farm labor (18 percent paid zero), compared to just 60 percent of the male maize farmers (40 percent paid zero); Figure 2. Women maize farmers paid an average of 83,000 FCFA (1FCFA = 0.001715 USD) to farm labor (comparable to the average 76,000 GNI/capita in Cameroon). Male maize farmers paid an average of 32,400 FCFA to farm labor, less than half the rate paid by female maize farmers. Indeed, over 39 percent of the female maize farmers paid at least 80,000 FCFA to labor, while only 11 percent of the male maize farmers paid 80,000 or more (Figure 2). Presumably, the male farmers enjoyed more unpaid family labor than their female counterparts.

**Table 1.** Profitability for young males and females Maize and Rice producer and marketers.

| | Maize | | | | Rice | | | |
|---|---|---|---|---|---|---|---|---|
| | Males | | Females | | Males | | Females | |
| | Mean | Standard Deviation | Mean | Standard Deviation | Mean | Standard Deviation | Mean | Standard Deviation |
| Cost of labor | 32,192.6 | 38,556.5 | 83,086.0 | 74,571.6 | 29,315.3 | 29,052.5 | 25,264.4 | 24,755.5 |
| Cost of seed | 4860.3 | 5289.3 | 3833.1 | 3037.3 | 8270.5 | 5730.6 | 8300.0 | 4251.9 |
| Cost of fertilizer | 62,631.0 | 60,714.2 | 43,812.1 | 21,057.9 | 80,321.4 | 58,501.0 | 77,771.4 | 57,325.9 |
| Cost of herbicides | 9257.7 | 10,170.5 | 8688.9 | 4618.2 | 4724.3 | 9196.2 | 3932.1 | 2375.8 |
| Postharvest loses | 1950.2 | 3709.4 | 2924.8 | 3062.5 | 413.9 | 3491.2 | 121.4 | 917.2 |
| Cost of treatment | 3354.8 | 5356.9 | 4461.1 | 3755.0 | 10.4 | 176.8 | 0.0 | 0.0 |
| Drying fee | 0.0 | 0.0 | 0.0 | 0.0 | 236.1 | 1788.4 | 0.0 | 0.0 |
| Warehouse rent | 1305.3 | 1794.1 | 2143.3 | 1850.2 | 201.4 | 2146.5 | 0.0 | 0.0 |
| Tax paid | 1223.5 | 1856.5 | 2046.5 | 1810.3 | 3618.8 | 4537.1 | 3712.1 | 4320.7 |
| Cost of packaging | 370.0 | 812.6 | 1105.1 | 1163.3 | 0.0 | 0.0 | 0.0 | 0.0 |
| Cost of transportation | 14,955.3 | 27,307.0 | 17,152.9 | 13,736.3 | 19,285.2 | 27,741.6 | 15,437.9 | 12,012.1 |
| Total variables cost per hectare | 132,544.9 | 99,407.8 | 169,990.2 | 94,417.2 | 146,402.5 | 107,745.2 | 134,539.4 | 90,323.5 |
| Total revenue per hectare | 345,008.1 | 412,865.6 | 331,296.2 | 157,345.0 | 389,777.8 | 284,487.6 | 377,146.4 | 258,632.8 |
| Gross margin per hectare | 212,463.2 | 359,921.5 | 161,306.0 | 127,425.8 | 243,375.2 | 196,060.8 | 242,607.0 | 196,215.0 |
| Return per FCFA invested | 1.9 | 2.4 | 1.5 | 1.8 | 2.0 | 1.6 | 2.2 | 2.1 |

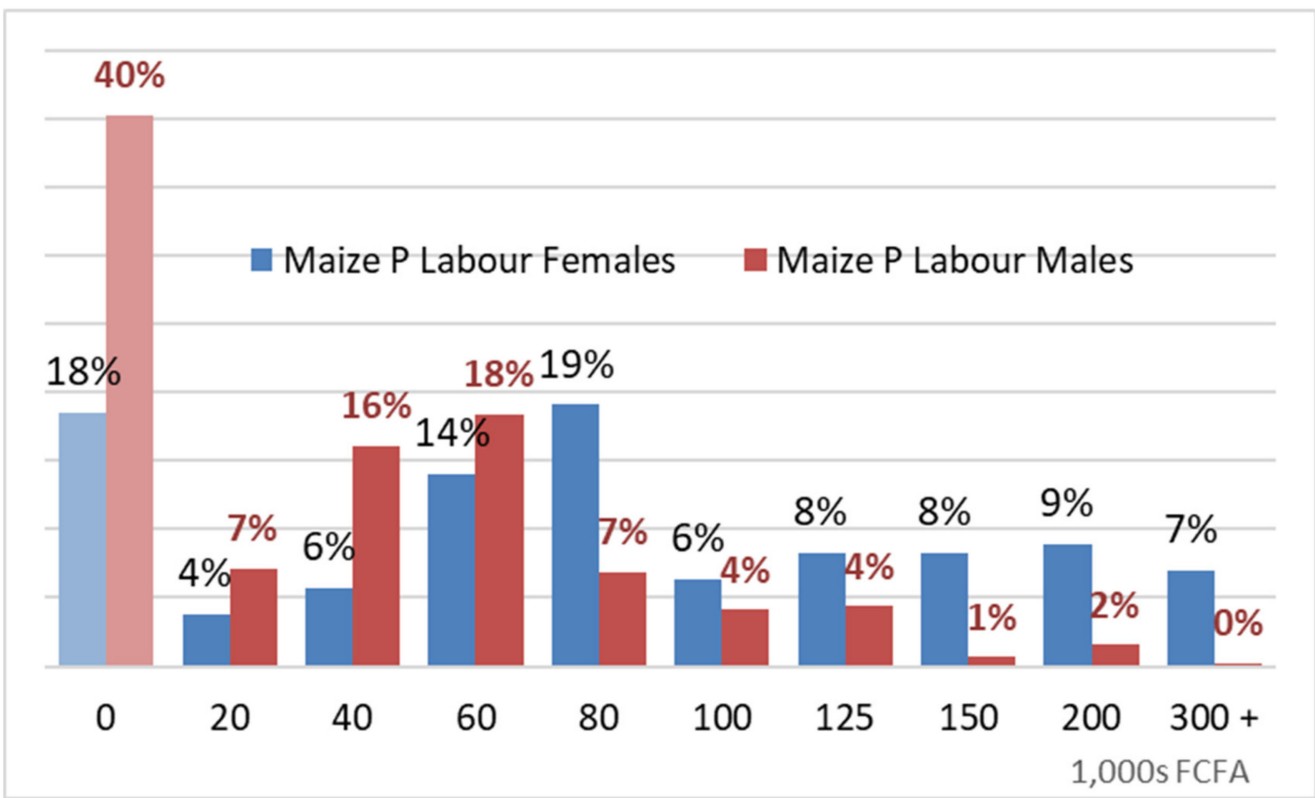

**Figure 1.** Percentage of young male and female maize farmers paying labor by rate paid.

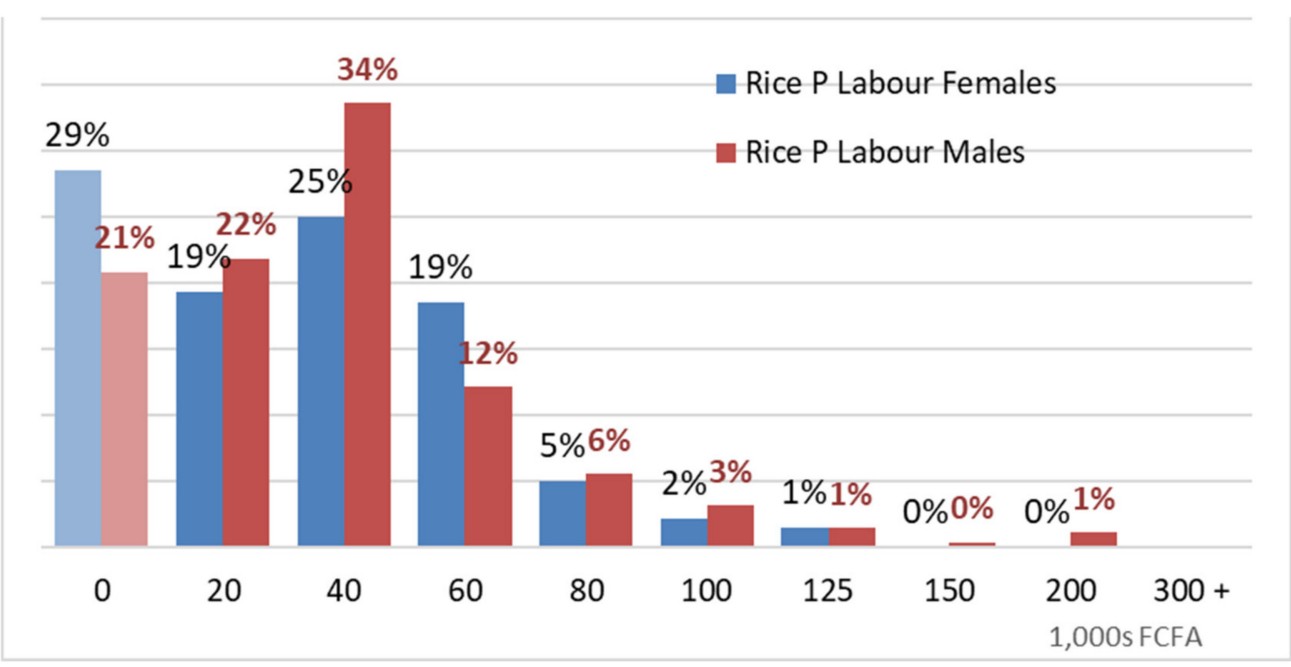

**Figure 2.** Percentage of young male and female rice farmers paying labor by rate paid.

Paying for more hired labor means contributing more to their local rural economies, but it does come at a personal cost. It means that female maize farmers took home a lower return than their male counterparts. Our survey data showed that female maize farmers averaged 1.5 FCFA per FCFA invested compared to 1.94 FCFA per FCFA invested among male maize farmers. The good news is that returns for both male and female maize farmers are positive and higher than the average rate of return per FCFA saved in a bank account at 5% interest rate. This finding is in line with Reference [30], who estimated the gender gaps in labor productivity to be 12 percent in sub-Saharan Africa.

Among rice producers and marketers, however, males were found to be more likely than females to pay labor (Figure 2). Seventy-nine percent of the male rice producers and marketers paid farm labor (21% paid zero), a larger share than the 71 percent of the female rice producers and marketers who paid farm labor (29 percent paid zero). Again, paying farm labor is a contribution to the rural economy, but a business cost. Male rice producer–marketers spent more on labor and enjoyed a lower average return on investment. The average return to males in our sample was 1.98 FCFA per FCFA invested compared to 2.20 FCFA for female rice producers and marketers in our sample.

Other differences between female and male agribusiness entrepreneurs, other than paid labor, credit, (Table 2), and prices received for rice, were not statistically significant differences.

**Table 2.** Percent of farmers obtaining credit.

|  | Maize | Rice |
|---|---|---|
| Young Males | 3% | 14% |
| Young Females | 1% | 6% |

Female producers and marketers sold their maize for about the same prices, on average, as their male counterparts (140–150 FCFA/kg). Figure 3 shows, however, that a larger share of the women (45 percent) sold maize for under 140 than men (32 percent). Indeed, 20 percent of the women sold for not more than 110, compared to 6 percent of the men. The same very small share of both male and female maize producers in our sample (1 percent) sold for 220.

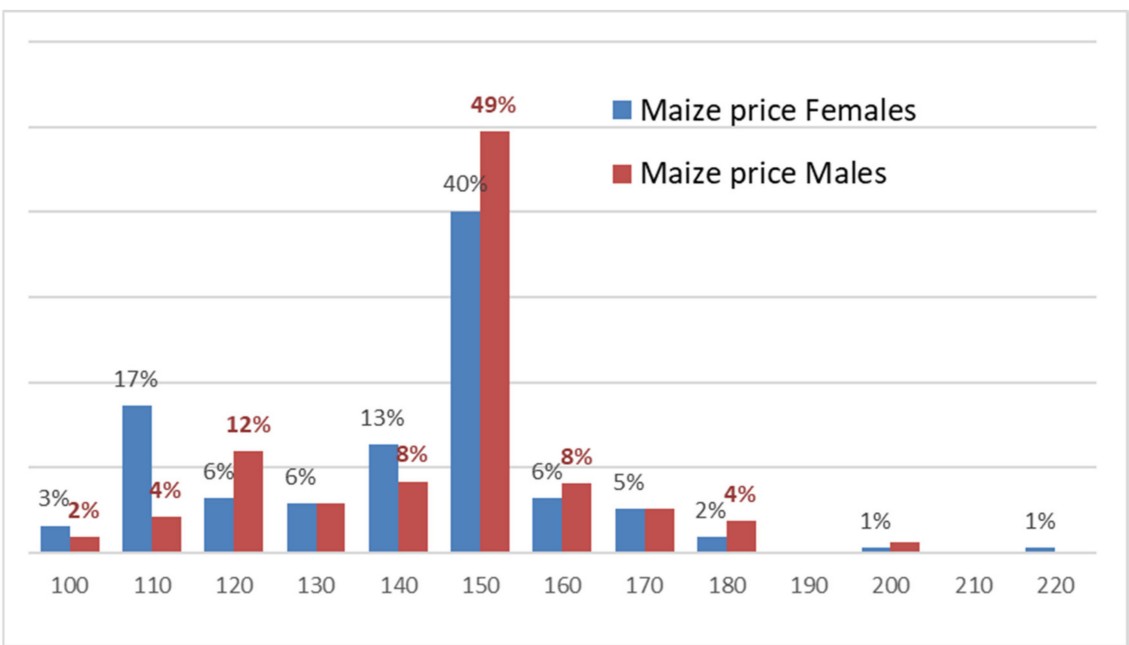

**Figure 3.** Percentages of young male and female maize farmers by sales price received.

Perhaps the clearest explanation for the higher return on investment enjoyed by female rice producers and marketers is the fact that a larger share of the females (33 percent) got a high price (200–220) for their rice than males (18 percent), Figure 4. As Figure 4 illustrates, the prices paid for rice in 2018 were bi-modal, with a third at 100–110 and a third at 200–220. The bi-modal nature of selling price is firstly due to the large quantity supplied of rice by the females during the harvest period which attracts a low selling price. Secondly, the limited access to capital makes female rice farmers store specific quantities to be sold later in the preparation of the subsequent cropping season, whereby there is a high demand for rice associated with an increase in selling price. The low price by young females compared to the young male counterparts could also follow References [21,37] who explained that females tend to be more risk-averse than males, they may prefer lower-risk/lower-return strategies.

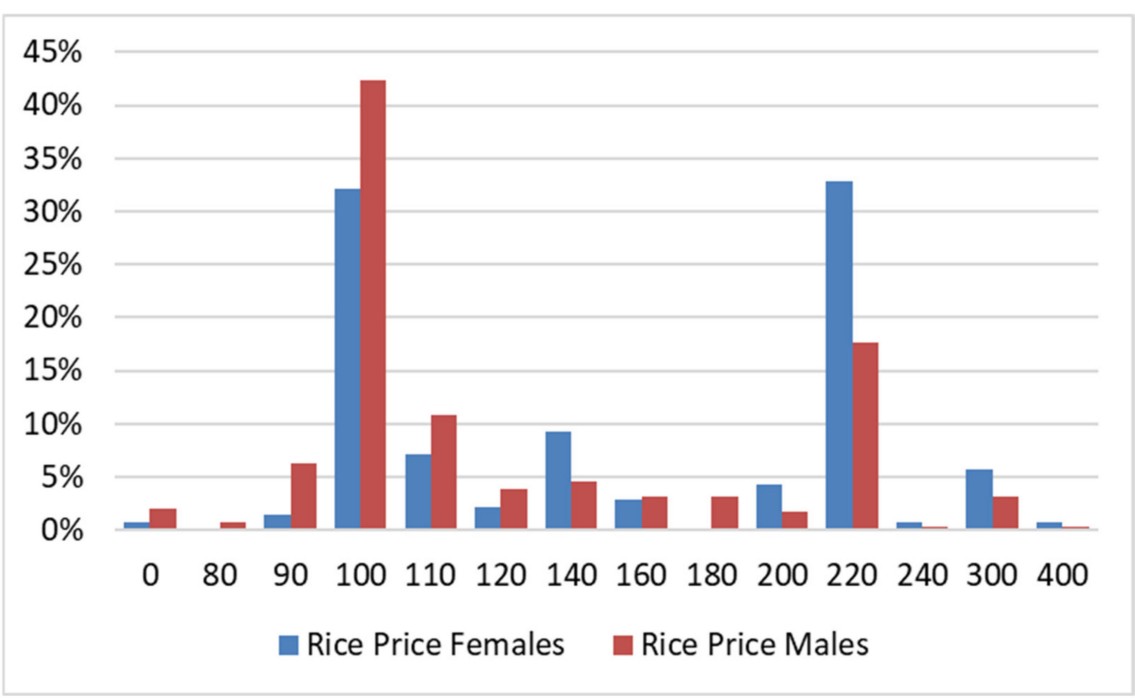

**Figure 4.** Percentages of young male and female rice farmers by sales price received.

Although few farmers in our sample obtained credit at all, a larger share of males got credit than females (Table 2), and rice farmers were more likely than maize farmers to obtain credit. Further study is warranted to determine if the relatively low rates reflect low credit demand or if they reflect low credit supply, credit rationing, or gender-bias. In line with this argument, Reference [47] argued that lenders discriminate against women indirectly because they prefer to lend to larger and more established firms. In addition, even if there is no overt discrimination, the banks could be rationally responding to women's disadvantaged backgrounds and endowment. For instance, Reference [48] noted, "Bank staff are not guilty of discrimination. Rather applicants' socialization and work-related experiences have disadvantaged them compared to male applicants."

Furthermore, the difference between the profitability of both groups in rice and maize marketing could be due to differences in the farm size (0.83 ha for males against 0.76 ha for females in rice marketing; 2.14 ha for males against 1.84 ha for females in maize marketing) and output harvested (4062.15 kg/ha for males against 3298.57 kg/ha for females in rice marketing; 3233.45 kg/ha for males against 2353.50 kg/ha for females in maize marketing). These results disagree with the findings of Reference [49] which found that female farmers have a higher profit in rice production than males in the Philippines. Moreover, this result is in line with the finding of Reference [50] that found that males have a higher gross margin in three focus crops (maize, rice, and soybean) in Northern Ghana. Furthermore,

Reference [51] explained that profit differences between males and females could be due to the large plots controlled by men in Ghana.

Farm sizes are similar. Women growing and marketing maize cultivated an average of 1.85 (2 ha) hectares, while the men cultivated an average of 2.07 (2 ha) hectares, which is not statistically significantly different. However, as Figure 4 shows, women do not operate larger maize farms. Only 3 percent of the females cultivated more than 3 hectares of maize, and the largest maize farm operated by a woman in our sample was 4 hectares. In contrast, 12 percent of the men cultivated more than 3 hectares of maize, and the largest maize area operated by a man in our sample was 40 hectares (Figure 5). This result is in line with Reference [52], which explained that there is ample evidence showing that in developing countries the distribution and control of land, property, and assets are skewed toward men and women's property rights are less secured. Thus, securing property rights over productive assets is important. Such property rights play a central role in investment decisions, allocative resources, and economic development [53]. Secondly, in many countries, social and sometimes legal norms require assets that are registered in a male spouse's name. For instance, Kantor [54] reports that women in South Asia have more limited ownership of assets and property that can be used as collateral due to both legal and traditional bars on female ownership.

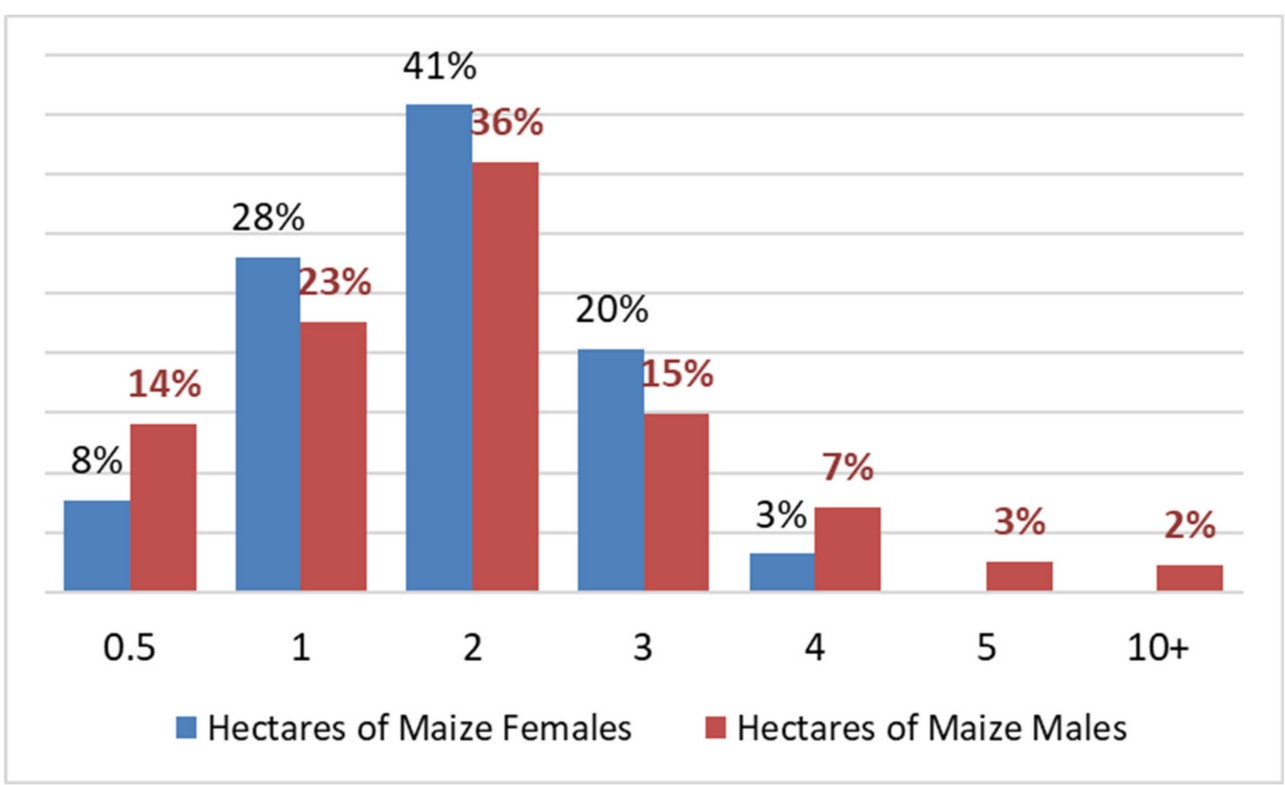

**Figure 5.** Percentages of young male and female maize producers by farm size.

The areas planted to rice are generally smaller than to maize, averaging 0.84 (0.8 ha) hectares for the males and 0.77 hectares (0.8 ha) among the female rice producer–marketers in our sample. Again, the difference in the average rice farm size is not statistically significant, but as Figure 6 shows, only men cultivated larger areas of rice.

In addition, the results of the profitability test for both researched groups show that those *p*-values are not significant when equal and non-equal variances are assumed for ??? as shown in Table 3. Therefore, the null hypothesis that states that there is no significant difference in profitability is accepted. This implies that a young female earns as much as the young male counterpart in rice production and marketing. Moreover, results show that p-values are significant when equal and non-equal variances are assumed for their

profitability. Therefore, the null hypothesis that states that there is no significant difference between their profitability is rejected. This implies that a young female earns less than the young male counterpart in maize production and marketing.

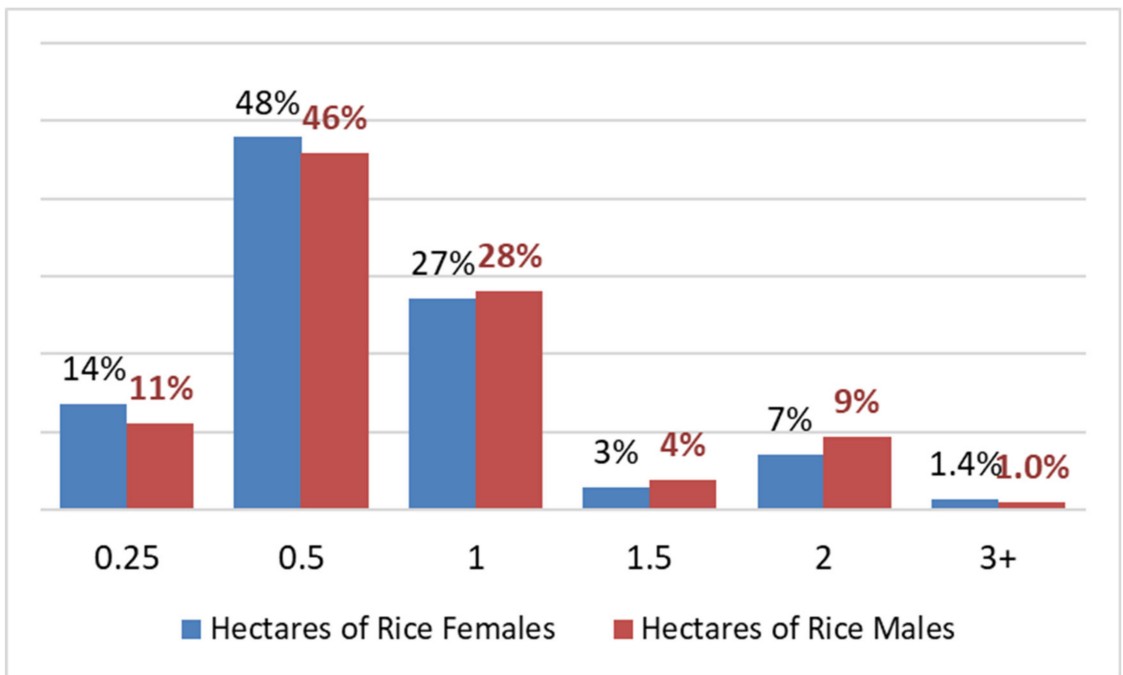

**Figure 6.** Percentages of young male and female rice producers by farm size.

**Table 3.** Test of difference between profitability, cost, and revenue for young males and females rice and maize producer and marketers in Cameroon.

| | Rice | | | Maize | | |
|---|---|---|---|---|---|---|
| **Profit** | **T Test** | **df** | **Significance (2-Tailed)** | **T Test** | **Df** | **Significance (2-Tailed)** |
| Equal variances assumed | 0.038 | 426 | 0.970 | 1.737 | 589 | 0.083 ** |
| Equal variances not assumed | 0.038 | 275.264 | 0.970 | 2.547 | 588.81 | 0.011 * |
| **Cost** | | | | | | |
| Equal variances assumed | 1.103 | 426 | 0.271 | −4.08 | 589 | 0.000 *** |
| Equal variances not assumed | 1.171 | 322.71 | 0.242 | −4.18 | 288.9 | 0.000 *** |
| **Revenue** | | | | | | |
| Equal variances assumed | 0.444 | 426 | 0.625 | 0.405 | 589 | 0.685 |
| Equal variances not assumed | 0.459 | 300.283 | 0.647 | 0.585 | 587.59 | 0.559 |

Source: Field Survey, 2018. N.B.***, ** and * are significant at 1%, 10%, and 5%, respectively.

### 4.3. Factors Influencing Young Male and Female Entrepreneurs' Participation in the Maize and Rice Agribusiness

Results show that age, household size, experience, years in school, cost of fertilizer, and selling price are the factors that influence participation in rice agribusiness; household size, years in school, monthly income, and cost of fertilizer are the factors that influence participation in maize agribusiness as shown in Table 4.

**Table 4.** Factors influencing young male and female entrepreneurs' participation in the maize and rice agribusiness.

| Variables | Rice | | Maize | |
|---|---|---|---|---|
| | Coefficient | T-Statistics | Coefficient | T-Statistics |
| Age | 0.04 ** | 2.12 | 0.006 | 0.51 |
| Household size | −0.07 ** | −2.02 | −0.06 ** | −2.10 |
| Experience | −0.07 *** | −2.79 | −0.02 | −1.09 |
| Membership | 0.39 | 1.37 | 0.12 | 0.52 |
| Years in school | −0.12 *** | −3.77 | −0.07 *** | −2.72 |
| Farm size | −0.28 | −0.98 | 0.11 | 0.81 |
| Monthly income | $-4.01 \times 10^{-6}$ | −0.71 | −0.00 * | −1.71 |
| Cost of seeds | −0.00 | −0.80 | −0.00 | −1.23 |
| Cost of fertilizer | $6.59 \times 10^{-6}$ ** | 2.02 | $-9.91 \times 10^{-6}$ ** | −2.24 |
| Cost of herbicides | −0.00 | −0.53 | $-1.87 \times 10^{-6}$ | −0.07 |
| Selling price | 0.006 ** | 2.26 | −0.009 | −1.60 |
| Distance to market | −0.001 | −0.21 | 0.01 *** | 3.34 |
| Constant | −1.03 | −1.32 | 1.84 | 1.84 |
| Log likelihood | −242.91 | | −298.61 | |
| Pseudo $R^2$ | 0.510 | | 0.527 | |
| LR $chi^2$(12) | 85.25 | | 87.02 | |
| Prob > $chi^2$ | 0.000 *** | | 0.000 *** | |

*Source*: Field survey, 2018. ***, ** and * are significant at 1%, 5% and 10% respectively.

Moreover, results show that LR $chi^2$ are, respectively, significant at a 1 percent level of probability for both researched groups. This indicates the goodness of fit of the model and the joint significance of all the variables used in it. Therefore, the hypothesis that stated that socioeconomic and financial factors have no significant influence on participation by young males and females in the marketing of rice and maize is rejected.

Specifically, the coefficient of age is positive and significant at a 5 percent level of probability. This result is contrary to *a priori* expectations and suggests that age increases the probability of both groups participating in rice agribusiness. This result could be due to the fact that aging farmers are accustomed to routine in their farm business and may not have other activities to keep them busy, which could lead to an increase in their participation in the marketing of rice. This result agrees with the findings of Reference [55] which found that age increases the probability of market participation among maize farmers in Ogbomoso zone, Oyo State, Nigeria.

The coefficient of household size is negative and significant at a 5 percent level of probability. This result is contrary to *a priori* expectations and implies that household size decreases the probability of youth participation in rice marketing.

Similarly, for maize, the coefficient of household size is negative and significant at 5 percent level. This result could be explained by the average size of household members (7 and 5 members for maize, although 7 and 6 members for rice) which could reduce the marketable quantity of rice and maize, given their small farm sizes (2.14 ha and 1.84 ha for maize; 0.83 ha and 0.7 ha for rice). This result is contrary to *a priori* expectations and suggests that increases in household size decrease the probability of youth participation in maize marketing. This result implies that most households do not produce more than their home consumption which will, therefore, reduce the quantity sold. These results agree with the findings of Reference [56] which found a decline in the Philippines in market participation among smallholder milk producers with an increase in the number of household members. They explained that farmers might not have the excess milk to participate in marketing as the number of family members increased [57].

The coefficient of experience is negative and significant at a 1 percent level of probability. This result is contrary to *a priori* expectations and suggests that experience reduces the probability of both groups participating in rice marketing. This could be explained by the fact that experienced farmers are reluctant to adopt new strategies based on the observed realities in rice marketing and are willing to maintain their old customs which will, there-

fore, incur losses in the business and reduce their participation. This result is contrary to the findings of Reference [58] which explains that farmers with more years of experience are likely to have a better understanding and more knowhow about appropriately managing their coffee farms under tough economic conditions than less experienced farmers.

The coefficient of years in school is negative and significant at a 1 percent level of probability. This result is contrary to *a priori* expectations and suggests that years in school reduce participation in rice marketing by youth entrepreneurs in both researched groups. Similarly, for maize, the coefficient of years in school is negative and significant at a 1 percent level of probability. This result is contrary to *a priori* expectations and suggests that years in school reduce participation by both groups in maize marketing. These results could be due to the fact that people with more education are more likely to look for jobs out of the agricultural sector from lack of incentives such as a farm input subsidy. This result is contrary to the finding of Reference [59], which found that education increases market participation among maize producers in Oyo State, Nigeria.

The coefficient of monthly income is negative and significant at 10 percent level of probability. This result is contrary to *a priori* expectations and suggests that increases in monthly income decrease participation by both groups in maize marketing. This result could be explained by Reference [60], which reports that fresh fruit farmers with high off-farm income are more likely not to participate in the market. This change could be attributed to diverse eating habits and a shift in priorities for those with higher income who, therefore, would sell more of their rice to acquire other foods or invest in real assets [61].

The coefficient for the cost of fertilizer for rice marketing is positive and significant at 5 percent level of probability. This result is contrary to *a priori* expectations and suggests that increase in the cost of fertilizer increases the probability of youth participation in rice marketing. This result could be due to the fact that the majority of participants in both researched groups are operating on a full-time basis and have no choice other than participation in the marketing of rice, given the increment in the cost of fertilizer. Furthermore, the increase in the cost of fertilizer may be translated into an increase in selling price which, in turn, will cover up all expenses and increase profits of both groups, thereby increasing their participation.

In contrast, for maize, the coefficient for the cost of fertilizer is negative and significant at 5 percent level of probability. This result is in line with *a priori* expectations and suggests that increases in the cost of fertilizer reduces participation by both groups in maize marketing. This could be explained by Reference [62], which argues that imperfections in fertilizer markets can limit the participation of smallholders in output markets. The rationale behind their analysis is that, in some contexts, transaction costs in fertilizer markets may rise and can even be higher than those in output markets; it results in low use of fertilizer, which mitigates the production of a marketable surplus [63].

The coefficient of the selling price is positive and significant at 5 percent level of probability. This result is in line with *a priori* expectations and suggests that increases in the selling price increase youth participation in rice marketing. This could be explained by the fact that the selling price serves as an incentive for farmers' income and more investment in the business and could, therefore, increase participation by both groups in rice marketing. This result is in line with Reference [57], which finds that increase in selling price increases the participation of smallholder farmers in red bean marketing in Halaba special district of Ethiopia.

Finally, the coefficient of distance to market is positive and significant at 1 percent level of probability. This result is contrary to *a priori* expectations and suggests that increase in distance to market increases participation by both groups in maize marketing. This is completely the reverse to past empirical findings in this area of agricultural market participation. Past empirical studies established that distance was inversely related not only to the decision to participate in the market but also to the amount or volume sold [64–66]. However, there could be a valid reason for this finding. Given the fact that maize is a staple crop, a bulky and low-value grain, farmers who are closer to main markets are likely to

grow high-value and more perishable crops for the nearby niche markets and only end up buying maize afterward. This is perfectly in line with the von-Thunen theoretical model of land use [65]. This result is in line with the findings of Reference [66], which state that an increase in distance to the market increases the probability of sales to the local traders and brokers in the case of banana marketing in Muranga County, Kenya.

However, for rice, the coefficients of membership, farm size, price of seeds and herbicides, and distance to market are not significant for both groups and have no significant influence. Similarly, for maize and both researched groups, the coefficients of age, experience, membership, farm size, cost of seeds and herbicides, and selling price are not significant and have no influence.

## 5. Conclusions

The challenge of rural un- and underemployment, especially among young women, can be solved in part if more young Cameroonians choose agribusiness entrepreneurship. Our research shows that young women can be as productive and successful as young men, especially in rice agribusiness. Moreover, women who pay more for hired labor in maize farming deserve the appreciation and respect of their local community and the country because they provide income opportunities to others even though it lowers their personal returns.

By these measures, maize and rice production and marketing in Cameroon appear to be good ways to earn a living, raise local food security, and/or promote development in rural communities, where nearly half the population lives. However, this study was limited to only two (2) cereals crops and suggestion that further research should include activities on cash and export crops as well as the sociological background of the youths should be included in the empirical analysis will bring a broader view on the performance of youth in agribusiness.

## 6. Policy Implications

i. Given that hired labor constitutes a high proportion in maize and rice production and marketing for young women, it is suggested that young female rice and maize producers and marketers should be encouraged to register into community association with the aim to set up community farm labor activities, which could considerably reduce the cost of labor and increase their performance.

ii. Given that young women faced differential in selling price compared to young male counterpart probably due to averse risk, it is important to set up policies which will include supporting and training programs specially designed to develop technical and managerial skills to build confidence in young women entrepreneurs.

iii. Moreover, there is a need for young women to expand their business connections with successful entrepreneurs for support and advice on how to manage price risk through gender unbiased based membership of association to remedy the networking opportunities. Following that young women are as productive as the counterpart young men, young women farmers should develop multiple avenues through which land can be accessed (e.g., church membership, cooperatives) so they can be less dependent on their husbands for land access.

iv. Incentives such as single-digit interest rates with no collateral security should be directed to young women to receive more credit for purchasing agrochemicals and improved varieties of seed.

**Author Contributions:** Conceptualization, D.C.R.F.; methodology, D.C.R.F., E.N.O. and M.S.; data curation, H.U.U., D.D.D. and E.N.O.; writing—original draft preparation, D.C.R.F., O.H.O. and A.T.N.; writing—review and editing, D.C.R.F., H.U.U., D.N.P.M. and M.G.A.; supervision, D.N.P.M. All authors have read and agreed to the published version of the manuscript.

**Funding:** The grant used for this study came from the International Fund for Agricultural Development (IFAD) through the International Institute of Tropical Agriculture (IITA) under the project "Youth Researching Youth: Competitive Fellowships for Young African Scholars Researching Youth Engagement in Rural Economic Activities in Africa.

**Data Availability Statement:** Not applicable.

**Acknowledgments:** We are grateful to the International Fund for Agricultural Development (IFAD) grant support and the International Institute of Tropical Agriculture (IITA)'s technical support. Any opinions, findings, and conclusions, or recommendations expressed in this material are those of the author(s) and do not necessarily reflect the views of the IFAD and the IITA.

**Conflicts of Interest:** The authors declare that they have no affiliations with or involvement in any organization or entity with any financial interest (such as honoraria; educational grants; participation in speakers' bureaus; membership, employment, consultancies, stock ownership, or other equity interest; and expert testimony or patent-licensing arrangements) or non-financial interest (such as personal or professional relationships, affiliations, knowledge or beliefs) in the subject matter or materials discussed in this manuscript.

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
