# Peer review of "Assessing the Performance and Participation among Young Male and Female Entrepreneurs in Agribusiness: A Case Study of the Rice and Maize Subsectors in Cameroon"

_sustainability, doi:10.3390/su13052690_

Round 1
Reviewer 1 Report
This paper considers the use of various inputs in maize/rice farming according to gender in Cameroon. It is an interesting and important topic, and I recommend the authors be given the opportunity to resubmit after revision.
I have no quarrel with the methodology or research design. The research is sound and well conducted. I recommend that the Discussion and Policy Recommendations be the main focus of the revision. Currently, the Policy Recommendations are motivated by the findings (which is good), but the actions recommended have little theoretical or empirical justification. For example, women and men price their products differently. To recommend a solution to this, why is a price control policy, which would generate significant deadweight loss, favored over an information campaign or training program that teaches farmers about the prices they can demand, a policy which would allow the market to fluctuate? As another example, for the third recommendation, what is the current land tenure system, and how could it "give more consideration to young women to acquire more land"?
A broader discussion about the ways in which these findings can illuminate important information to scholars who do not focus on Cameroon, but do focus on agribusiness, would help draw in references to policies that have been successful in other places. These references could then inform the recommendations for Cameroon.
As long as the recommendations remain weakly supported, which they do in the current version, I cannot recommend publication. With further development of the justification for the particular recommendations given, or with a revision of the recommendations themselves to more carefully consider the likely implications and unintended consequences of the policies recommended, this article could be quite worthy of publication in the journal.
Author Response
Point 1: The reviewer I was mainly concerned with our policy recommendations II and III, we were able to go through literature and acknowledge his/her view following the issue of risk that could influence young women to sell their farm product lower than the male counterparts. The following recommendations were therefore suggested for policy advocacy in page 23 to handle the issue of price and land acquisition by women
Response 1
- Given that young women faced differential in selling price comparatively to young male counterpart probably due to averse risk, there is to set up policies which will include supporting training programs especially designed to develop a technical and managerial skills to build confidence on young women entrepreneurs
- Also, there is need for young women to expand their business connections with successful entrepreneurs for support and advice on how to manage price risk through gender unbiased based membership of association to remedy the networking opportunities.
- Following that young women are as productive as the counterpart young men; young women farmers should develop multiple avenues through which to access land (e.g., church membership, cooperatives) so they can be less dependent on their husbands for land access.
Point 2: our findings needed back up from related findings
Response 2
Our results was also back up with related findings on gender and entrepreneurship on pages 10, 11, 12, 14
Page 10 (This result could further be explained by [36] who found that although profits are significantly higher for male-controlled small and micro enterprises than female-controlled small and micro enterprises due to risk (i.e. the variation in profits). Also, [34] found that while returns to capital in female-owned firms are significantly lower than in male-owned firms, the returns to time and hours of labor are actually higher for female-run firms. This suggests that economic outcomes can be improved if women could devote more time to their business ventures)
Page 11 (This finding is in line with [30] who estimated the gender gaps in labor productivity to be 12 percent in Sub-Saharan Africa).
Page 12 (The low price sold by young females compared to the young male counterparts could also follow [37] and [21] who explained that females tend to be more risk averse than males, they may prefer lower-risk/lower return strategies)
Page 13 (In line with this argument, [47] argued that lenders discriminate against women indirectly because they prefer to lend to larger and more established firms. In addition, even if there is no overt discrimination, the banks could be rationally responding to women’s disadvantaged background and endowment. For instance, [48] noted, “Bank staff are not guilty of discrimination. Rather applicants’ socialization and work-related experiences have disadvantaged them compared to male applicants)
Page 14 (This result is in line with [52] who explained that there is ample evidence showing that in developing countries the distribution and control of land, property and assets is skewed toward men, and women’s property rights are less secured. Thus, securing property rights over productive assets is important. Such property rights play a central role in investment decisions, allocative resources, and economic development [53]. Second, in many countries social and sometimes legal norms require assets are registered in a male spouse’s name. For instance, Kantor [54] reports that women in South Asia have more limited ownership of assets and property that can be used as collateral due to both legal and traditional bars on female ownership)

Reviewer 2 Report
More literature on gender entrepreneurship needed.
Cite and discuss:
Gender entrepreneurship and global marketing, Journal of Global Marketing 2017
Gender and Family Entrepreneurship, Routledge, 2017
Author Response
Point 1: literature review should be added
Response 1: A literature review on gender and entrepreneurship was added in the article on pages 4 and 5. Our literature development focused on gender and entrepreneurship. We contend that gender and entrepreneurship could be assessed in the following ways: economic and non-economic outcomes [21]. Studies in entrepreneurship have invoked a variety of theoretical perspectives to explain differences between female-owned businesses and male-owned businesses [22; 23]
Non-economic outcomes
Following [21] non-economic outcomes such as self-empowerment, time flexibility, status in the community, satisfaction with life, and work-life balance are more important for women than for men. As we discuss below, the narrow definition of success that highlights only economic motivations for entering entrepreneurship tends to better fit the male model. Women often have different motivations when they enter self-employment and they evaluate the success of their business using different metrics than men. Thus, in discussing female entrepreneurial performance, it is important to include non-economic outcomes, which are frequently the driving forces behind women’s choice of self-employment. However, the literature on non-economic business outcomes of women entrepreneurs is very sparse. There is some limited evidence that in evaluating their firm’s performance, women tend to focus more on non-economic factors, such as personal fulfillment, flexibility and desire to serve the community [24]. In Sweden, [25] found that while women entrepreneurs were similar to men in their pursuit of economic goals, women also valued other goals, including customer satisfaction and personal flexibility. In a study of Lebanese female entrepreneurs, [26] found that many women named non-financial aspects of their businesses, such as love of what they do every day and rendering an important service to the community, as important satisfying factors. A study of U.S. entrepreneurs revealed that women were more likely than men to develop strategies that emphasized product quality and less likely to emphasize cost efficiency [27].
Economic Outcomes
Many studies find that female-owned enterprises exhibit lower profitability and productivity than male owned ones. The differences vary widely across studies [21]. [28] find that female‐owned firms throughout Latin America tend to be smaller than male‐owned firms in terms of sales and number of employees. Similarly, [29] find that female-owned firms in Sub‐Saharan Africa have sales that are 31 percent lower than male‐owned firms. There could be many reasons why female-run businesses are smaller. In Sub Saharan Africa, [30] estimate the gender gaps in labor productivity to be 12 percent. [31] analyze rural non‐farm entrepreneurship in Ethiopia and find that male-owned firms are three times more productive than female‐owned ones. Some of the differences in performance can be explained by the type of firms women operate. In particular, the size and sector of the firm often explain a large portion of the differences in performance. For example, in the U.S. [32] estimates that women’s concentration in the personal services sector explains as much as 14 percent of the earnings differential. [33] find that once differences in sector are accounted for; there is no longer a significant difference in performance between male and female-owned businesses. However, many studies find that even after controlling for firm characteristics, there are still differences in performance. For example, in Madagascar, [34] find that the estimated gender performance gap in value added is 28 percent even after controlling for factor inputs endowment, sectors and the owner's human capital. In Sub-Saharan Africa, only about one-third of the productivity gap is explained by differences in the types of businesses women run: smaller firms, firms that are unaffiliated with other businesses and firms that are not registered [30]. In Uganda, a small sample mixed-methods study by [35] find that when women cross over into male-dominated sectors, they attain higher returns than women in female-dominated sectors. In other words, the returns in male-dominated sectors are high not only for men. Even if women get lower profits than men they are still making more than in female-dominated sectors.
An important aspect of performance evaluation that received little attention is the risk-return trade-off. Because women tend to be more risk averse than men, they may choose to focus on lower risk/lower return strategies, rather than high risk/high return strategies [21]. [36] find that although profits are significantly higher for male-controlled small and micro enterprises, so is the risk (i.e. the variation in profits). [37] argue that it is inappropriate to compare returns from these different types of businesses without considering the differences in risk. They posit that inadequate control for differences in risk may explain why most of the previous literature observed differences in performance between male and female businesses. Indeed, when they control for risk adjusted returns using the Sharpe ratio, measured as the ratio of profits over the variance of the profits, they find no differences in performance in their U.S. sample.

Reviewer 3 Report
Abstract
- The abstract covered relevant aspects expected for a good abstract. It provided a succinct account of the orientation to the problem, objectives, methods, results and practical implications. However, the abstract omitted how the data was analysed and the contribution of the study.
Introduction
- The author should introduce the writers to the concept of Agribusiness, young male entrepreneurs and young female entrepreneurs.
- The author was silent on the case for rice and maize sub-sectors in Cameroon. The author should motivate for the case for rice and maize sub-sectors in Cameroon.
- The author should add the outline of other sub-sections in the last paragraph of the introduction after the aim and objectives of the study.
Literature Review
- There is no literature review. The author should add a sub-section on literature review.
Methodology
- The author should provide a justification for the study area. Why was the study conducted in Cameroon?
- The author should include the research philosophy, research design and research approach and provide a justification for the methodology.
- The author should mention how reliability and validity were ensured for the study.
- The data analysis sub-section is not detailed.
Results and Discussion
- The results were presented in a logical sequence and are linked to the aim and objectives of the study. However, the discussion should also explain the strengths and limitations of the methods.
- The author should clearly state the research hypotheses for the study. I only noticed the decision of hypotheses in the findings.
- It is important for the author to also explain how the key findings relate to previous research.
Conclusion
- Incorporate the limitation of the study and avenue for future research.
References
- The references are relevant and current. 25 out of the 33 references (75%) are within the time line of 10 years.
Author Response
Abstract
Point 1: The abstract covered relevant aspects expected for a good abstract. It provided a succinct account of the orientation to the problem, objectives, methods, results and practical implications. However, the abstract omitted how the data was analysed and the contribution of the study.
Response 1: We added the contribution and the data analysis techniques of the study on page 1 (The data were analysed using gross margin, student t-test and logistic regression. Our study contributes to the literature by showing that young women agribusiness entrepreneurs are as competitive as the male counterparts and that the opportunities for young women in agriculture are good).
Introduction
Point 1: The author should introduce the writers to the concept of Agribusiness, young male entrepreneurs and young female entrepreneurs.
Response 1: The concept of Agribusiness, young male entrepreneurs and young female entrepreneurs were added on page 2. (Agribusiness encompasses a wide range of activities that generate economic value. Agribusiness is comprised of not only farming, but all the other industries and services that connect farmers to consumers. It is traditionally thought that as an economy develops the role of agriculture in the economy’s GDP and employment rates decreases. This trend has been observed in Sub-Saharan Africa where agriculture’s contribution to GDP has fallen from 43 percent in 1965 to 12 percent in 2008. If agricultural activity continues to be limited to crop and livestock production, it will fail to contribute to output growth and poverty reduction [2]. However, agriculture’s contribution could be significantly enhanced by strengthening linkages with industry through agro-processing and providing value-addition to agricultural products, as well as improving post-harvest operations, storage, distribution and logistics. Such an agribusiness development path paves the way for economic growth, structural transformation and improved technical skills which in turn can catalyze economic activities and connect major economic sectors [2]).
(Entrepreneurial activities play a critical role in the development and well-being of societies [3; 4]. Thus, various stakeholders including governments, non-profits, researchers, and individuals are interested in facilitating the development of supportive entrepreneurial ecosystems. However, growth of young female entrepreneurship has lagged those of men in many developed and in most developing countries. Understanding potential road-blocks that female entrepreneurs faced are important for increasing their participation in the entrepreneurial activity [4]).
Point 2: The author was silent on the case for rice and maize sub-sectors in Cameroon. The author should motivate for the case for rice and maize sub-sectors in Cameroon.
Response 2: We motivated for the case for rice and maize sub-sectors in Cameroon on pages 2 and 3 (In Cameroon, maize was mainly considered a food crop in past decades and not as a cash crop [6]. However, with high demands from brewery companies and the livestock sector, production has increased its importance as a cash crop. Maize production has increased drastically from 650000 tons (t) in 2000 to 1,647,036 t in 2013. Rice has become the most rapidly growing food source for millions of people in Cameroon [6]. Accordingly, the Government of Cameroon established three development companies [Société d’ Expansion et de Modernisation de Riziculture de Yagoua (SEMRY) in 1954; Upper Noun Valley Development Authority (UNVDA) in 1974; and Société de Dévelopement de la Riziculture dans la plaine de Mbo (SODERIM) in 1978] to boost rice production and the farmers’ ability to increase their earnings (profitability). Despite these efforts, Cameroon produces an estimated 80,000 t of rice annually which is far short of the over 500, 000 t required to meet national demand [7; 8])
Point 3: The author should add the outline of other sub-sections in the last paragraph of the introduction after the aim and objectives of the study.
Response 3: We added the outline of other sub-sections in the last paragraph of the introduction after the aim and objectives of the study on page 4 (The paper is structured as follows: Section 1 presents a brief/research hypothesis. Section 2 shows a literature review. Section 3 elaborates the methodology which includes the study area, method of data collection, population and sampling techniques, techniques of data analysis, validity and reliability of data, and models specification. Section 4 presents the results and discussion; the last section concludes the article).
Literature Review
Point 1: There is no literature review. The author should add a sub-section on literature review.
Response 1: A literature review on gender and entrepreneurship was added in the article on pages 4 and 5. Our literature development focused on gender and entrepreneurship. We contend that gender and entrepreneurship could be assessed in the following ways: economic and non-economic outcomes [21]. Studies in entrepreneurship have invoked a variety of theoretical perspectives to explain differences between female-owned businesses and male-owned businesses [22; 23]
Non-economic outcomes
Following [21] non-economic outcomes such as self-empowerment, time flexibility, status in the community, satisfaction with life, and work-life balance are more important for women than for men. As we discuss below, the narrow definition of success that highlights only economic motivations for entering entrepreneurship tends to better fit the male model. Women often have different motivations when they enter self-employment and they evaluate the success of their business using different metrics than men. Thus, in discussing female entrepreneurial performance, it is important to include non-economic outcomes, which are frequently the driving forces behind women’s choice of self-employment. However, the literature on non-economic business outcomes of women entrepreneurs is very sparse. There is some limited evidence that in evaluating their firm’s performance, women tend to focus more on non-economic factors, such as personal fulfillment, flexibility and desire to serve the community [24]. In Sweden, [25] found that while women entrepreneurs were similar to men in their pursuit of economic goals, women also valued other goals, including customer satisfaction and personal flexibility. In a study of Lebanese female entrepreneurs, [26] found that many women named non-financial aspects of their businesses, such as love of what they do every day and rendering an important service to the community, as important satisfying factors. A study of U.S. entrepreneurs revealed that women were more likely than men to develop strategies that emphasized product quality and less likely to emphasize cost efficiency [27].
Economic Outcomes
Many studies find that female-owned enterprises exhibit lower profitability and productivity than male owned ones. The differences vary widely across studies [21]. [28] find that female‐owned firms throughout Latin America tend to be smaller than male‐owned firms in terms of sales and number of employees. Similarly, [29] find that female-owned firms in Sub‐Saharan Africa have sales that are 31 percent lower than male‐owned firms. There could be many reasons why female-run businesses are smaller. In Sub Saharan Africa, [30] estimate the gender gaps in labor productivity to be 12 percent. [31] analyze rural non‐farm entrepreneurship in Ethiopia and find that male-owned firms are three times more productive than female‐owned ones. Some of the differences in performance can be explained by the type of firms women operate. In particular, the size and sector of the firm often explain a large portion of the differences in performance. For example, in the U.S. [32] estimates that women’s concentration in the personal services sector explains as much as 14 percent of the earnings differential. [33] find that once differences in sector are accounted for; there is no longer a significant difference in performance between male and female-owned businesses. However, many studies find that even after controlling for firm characteristics, there are still differences in performance. For example, in Madagascar, [34] find that the estimated gender performance gap in value added is 28 percent even after controlling for factor inputs endowment, sectors and the owner's human capital. In Sub-Saharan Africa, only about one-third of the productivity gap is explained by differences in the types of businesses women run: smaller firms, firms that are unaffiliated with other businesses and firms that are not registered [30]. In Uganda, a small sample mixed-methods study by [35] find that when women cross over into male-dominated sectors, they attain higher returns than women in female-dominated sectors. In other words, the returns in male-dominated sectors are high not only for men. Even if women get lower profits than men they are still making more than in female-dominated sectors.
An important aspect of performance evaluation that received little attention is the risk-return trade-off. Because women tend to be more risk averse than men, they may choose to focus on lower risk/lower return strategies, rather than high risk/high return strategies [21]. [36] find that although profits are significantly higher for male-controlled small and micro enterprises, so is the risk (i.e. the variation in profits). [37] argue that it is inappropriate to compare returns from these different types of businesses without considering the differences in risk. They posit that inadequate control for differences in risk may explain why most of the previous literature observed differences in performance between male and female businesses. Indeed, when they control for risk adjusted returns using the Sharpe ratio, measured as the ratio of profits over the variance of the profits, they find no differences in performance in their U.S. sample.
Methodology
Point 1: The author should provide a justification for the study area. Why was the study conducted in Cameroon?
Response 1: We provided a justification on why was the study conducted in Cameroon on page 6 (Two challenges and an opportunity motivate to carry out this study in Cameroon. The challenges are rural un- and underemployment and poverty. In 2014, young women were seventy percent (467,700 young women) of the youths aged 15-24 years in Cameroon not in employment, education or training and a third of all households were officially poor rural underemployment and poverty despite the fact that farming accounts for 70% of the workforce but contributing for just 17% of GDP [38]. Half a million young women and a quarter million young men ready and willing to grow food, make goods, and provide services to others. It is a huge resource. How to mobilize them?
Point 2: The author should include the research philosophy, research design and research approach and provide a justification for the methodology.
Response 2: The research philosophy, research design and research approach and provide a justification for the methodology were provided in page 6 (This study adopted an analytic form of survey which made use of cross-sectional data. This approach aims at determining the performance of young entrepreneurs engaged in maize and rice agribusiness. This approach examines the relationship between the revenue and cost incurred by young entrepreneurs as they exist in a defined population at a single point in time or over a short period of time).
Point 3: The author should mention how reliability and validity were ensured for the study.
Response 3: The reliability and validity of instruments are shown in the study on page 6 (Validity and reliability of instruments The research instrument was validated by pilot testing and by passing it through my supervisors, to ensure that it possesses both face and content validity. The reliability of the instrument was conducted using a test-retest method. In doing this, twenty (20) questionnaires were administered twice to two communities drawn from the sample frame within the interval of two weeks. The scores obtained were correlated using Pearson Product moment correlation coefficient(r) for scores obtained at the interval level, while the Spearman’s rank (rho) correlation was used for scores obtained at the ordinal level. A mean product-moment correlation coefficient (r) of 0.83 indicated high reliability).
Point 4: the data analysis sub-section is not detailed.
Response 4: The data analysis sub-section is detailed on pages 7 and 8 (Models Specification
Gross margin
Gross margin is one of the oldest and simplest analytical tools used in farm management. It has been used in a number of economic studies for analyzing the profitability of farm practice where fixed costs such as machinery, buildings, and other implements are not accountable [40].
Following [40] Gross Margin is given as:
Where,
Gross Margin is measured in FCFA/kg (1 FCFA = 0.0017 USD);
Selling Price is measured in FCFA; it is a price at which producers and marketers sell their produce (rice and maize) in the market. It is used because agricultural price fluctuations determine the interaction between supply and demand forces [41].
Quantity of output is measure in kg; it is the quantity of rice or maize sold by a producer-marketer. It is among common factors affecting the income of farmers [42].
Cost of farm labor is measured in FCFA. It is the cost incurred by rice and maize producers and marketers per laborer per day for activities such as clearing, planting, weeding, agrochemical application, and harvesting.
Cost of seeds is measured in FCFA; seeds are the primary input used by farmers, contributing at least 40% to the formation of crop yields [43]
Cost of artificial fertilizer is measured in FCFA; artificial fertilizer is a substance containing the chemical elements that improve the growth and productiveness of plants.
Cost of herbicides is measured in FCFA. Their use is due to the availability of low-cost herbicides and because of a similar shortage of manpower [44]
Postharvest processing is measured in FCFA. It is the sum of the quantity of outputs lost in operations such as threshing, transportation, processing, storage, and exchange before they reach the consumer
Cost of chemicals is measured in FCFA. It represents the chemicals used for the preservation of rice and maize after the harvest.
Drying fee is measured in FCFA. It is the amount paid off-farm to laborers because grain to be stored through warm weather needs to be dry, but energy is needed to remove moisture.
Cost of packaging is measured in FCFA; packaging is an essential part of a long-term incremental development process to reduce losses [45].
Transportation cost is measured in FCFA; it is the cost incurred by rice and maize marketers by moving their goods to market.
Tax paid is measured in FCFA; it is a mandatory charge for everyone involved in income- generating activities in Cameroon.
Storage cost is the amount paid daily by rice and maize marketers for keeping their goods in a warehouse.
t-test Analysis
It was used to determine whether there is significant difference between the means of the two sets of samples (young male and young female) drawn from the same sample frame. The respondents are tested under the same period.
The t- statistic to test whether the means (gross margins) are different will be calculated as:
t = (X1-X2)
√ (S21/n1) + (S22/n2)
Where;
X1 = mean value for credit users
X2 = mean value for non-credit users.
S21 and S22 are the sample variance for sample n1 and n2 respectively.
t, follows the distribution with n1+n2 – 2 degrees of freedom.
Results and Discussion
Point 1: the discussion should also explain the strengths and limitations of the methods.
Response 1: We explained the strengths and limitations of the methods used in the study on page 10 (Strengths and weaknesses of models used in this study
Gross margin has been used in a number of economic studies for analyzing the profitability of farm practice where fixed costs such as machinery, buildings, and other implements are not accountable. However, gross margin measures only the profitability of the firm and, ignores other factors such as an increase in the cost of production to secure a supplier or decrease in the selling price to increase market share etc. Gross profit may produce misleading figures of profit. In logistic regression outputs have a nice probabilistic interpretation, and the algorithm can be regularized to avoid over-fitting. Logistic models can be updated easily with new data using stochastic gradient descent. However, logistic regression tends to underperform when there are multiple or non-linear decision boundaries).
Point 2: The author should clearly state the research hypotheses for the study. I only noticed the decision of hypotheses in the findings.
Response 2: We stated the research hypothesis for the study on page 4 (H01: Is there a significant difference between the profit among young male and female entrepreneurs in maize and rice agribusiness)
Point 3: It is important for the author to also explain how the key findings relate to previous research.
Response 3: We also explain how the key findings relate to previous research on pages 10, 11, 12, 14
Page 10 (This result could further be explained by [36] who found that although profits are significantly higher for male-controlled small and micro enterprises than female-controlled small and micro enterprises due to risk (i.e. the variation in profits). Also, [34] found that while returns to capital in female-owned firms are significantly lower than in male-owned firms, the returns to time and hours of labor are actually higher for female-run firms. This suggests that economic outcomes can be improved if women could devote more time to their business ventures)
Page 11 (This finding is in line with [30] who estimated the gender gaps in labor productivity to be 12 percent in Sub-Saharan Africa).
Page 12 (The low price sold by young females compared to the young male counterparts could also follow [37] and [21] who explained that females tend to be more risk averse than males, they may prefer lower-risk/lower return strategies)
Page 13 (In line with this argument, [47] argued that lenders discriminate against women indirectly because they prefer to lend to larger and more established firms. In addition, even if there is no overt discrimination, the banks could be rationally responding to women’s disadvantaged background and endowment. For instance, [48] noted, “Bank staff are not guilty of discrimination. Rather applicants’ socialization and work-related experiences have disadvantaged them compared to male applicants)
Page 14 (This result is in line with [52] who explained that there is ample evidence showing that in developing countries the distribution and control of land, property and assets is skewed toward men, and women’s property rights are less secured. Thus, securing property rights over productive assets is important. Such property rights play a central role in investment decisions, allocative resources, and economic development [53]. Second, in many countries social and sometimes legal norms require assets are registered in a male spouse’s name. For instance, Kantor [54] reports that women in South Asia have more limited ownership of assets and property that can be used as collateral due to both legal and traditional bars on female ownership)
Conclusion
Point 1: Incorporate the limitation of the study and avenue for future research.
Response 1: We incorporated the limitation of the study and avenue for future research on page 23. However, this study was limited to only two (2) cereals crops and suggests that further research should include activities on cash and export crops as well as sociological background of the youths should be included in the empirical analysis which will bring a broad view on the performance of youth in agribusiness.

Round 2
Reviewer 1 Report
The authors have addressed my primary concerns. They included a much stronger foundation for the recommendations they offer, and the recommendations now flow more logically from their work.
There are some issues with the style and tone of the language that could be helped with editorial overview. Otherwise, I think this article should be accepted!
Reviewer 3 Report
I am satisfied with the changes effected by the author.